# COVID-19 Anti-Vaccine Sentiments in Malaysia: Narratives of Comments from Facebook Post

**DOI:** 10.3390/vaccines11040834

**Published:** 2023-04-13

**Authors:** Li Ping Wong, Haridah Alias, Yee Lian Wong, Megat Mohamad Amirul Amzar Megat Hashim, Yulan Lin, Zhijian Hu

**Affiliations:** 1Centre for Epidemiology and Evidence-Based Practice, Department of Social and Preventive Medicine, Faculty of Medicine, Universiti Malaya, Kuala Lumpur 50603, Malaysia; 2Department of Epidemiology and Health Statistics, School of Public Health, Fujian Medical University, Fuzhou 350122, China; 3Jepak Health Clinic, Kampung Baru Sebuan Besar, Bintulu 97000, Sarawak, Malaysia; 4Department of Primary Care Medicine, Faculty of Medicine, Universiti Malaya, Kuala Lumpur 50603, Malaysia

**Keywords:** antivaccine, COVID-19, social media

## Abstract

The anti-vaccination movement was an ongoing issue in Malaysia, a Muslim-majority country, even before the COVID-19 pandemic. It is unclear whether the introduction of new COVID-19 vaccines would similarly provoke anti-vaccine sentiments. This study analyzed COVID-19 anti-vaccine sentiments in the Malaysian community. Anti-vaccine comments from Facebook page posts were extracted. The qualitative software QSR-NVivo 10 was used to manage, code and analyze the data. The fast-track COVID-19 vaccine evoked the fear of unknown long-term effects, safety, effectiveness and the duration of protection. The *halal* status of the COVID-19 vaccines is important. Although it is permissible to use vaccines that are not certified *halal* under the state of *darurah* (emergency), there was doubt that the current state has reached the stage of *darurah* that warrants the use of vaccines. COVID-19 vaccine microchip conspiracy theories were raised. COVID-19 is viewed as only severe for vulnerable populations, and hence vaccination is not needed for the healthy. There were opinions that coronavirus treatments would be more beneficial than vaccination. The anti-COVID-19 vaccine sentiments uncovered in this study provide important insights for the formulation of public health messages to instill confidence in new COVID-19 vaccines. Despite the pandemic being nearly over and many people worldwide having received COVID-19 vaccines, the findings provide important insight into potential issues regarding the introduction of new vaccines in the event of future pandemics.

## 1. Introduction

The emergence of severe acute respiratory syndrome coronavirus 2 (SARS-CoV-2) in China in 2019, which rapidly spread across China and many other countries around the world, has caused an unprecedented shock to the global public health system and the world’s economy. Malaysia, a Muslim-majority country in the South-East Asia region, was equally greatly impacted by the COVID-19 outbreak. Malaysia announced the first case of SARS-CoV-2 infection on 25 January 2020. By the end of December 2020, Malaysia had a total of 103,900 cumulative confirmed coronavirus cases [1]. To date, no specific treatments for COVID-19 are available, and vaccination against the coronavirus is likely to be the most effective approach to containing the pandemic. The global rollout of the COVID-19 vaccines may likely end the COVID-19 pandemic [2]. Of significant importance to pandemic control, an unprecedented research effort has resulted in the rapid development of the vaccine against the coronavirus. The development of vaccines started as soon as the onset of the novel coronavirus outbreak first emerged in Wuhan, China in 2019. On 11 December 2020, the Food and Drug Administration (FDA) authorized the first COVID-19 vaccine for emergency use.

Vaccine hesitancy is a growing threat to global health security, particularly in the era of the COVID-19 pandemic. Tracking COVID-19 vaccine hesitancy is crucial to understanding any concerns about the vaccines. Often, anti-vaccine viewpoints are widespread in social media [3]. As global social media usage continues to grow, the potential to harness data generated from social media users has also grown [4]. The activities of social media users, through commenting or messaging, serve as new research avenues with which to harvest empirical insights into public opinion and responses. In Malaysia, many take to social media to voice concerns over COVID-19 vaccination before it was introduced [5]. It is important to investigate public opinion regarding the release of new vaccines. Therefore, we examined Malaysian public opinion via social media following the press releases that COVID-19 vaccines will soon be made available to the Malaysian public. Although many people worldwide have received the COVID-19 vaccine and COVID-19 cases and deaths continue to drop in most countries in the world, anti-vaccine sentiment captured during the early phase of the pandemic provides insight into issues surrounding the introduction of new vaccines in the event of future pandemics, particularly in Muslim-majority countries.

## 2. Materials and Methods

### 2.1. Data Collection

Researcher (MMAHMH), a vaccine advocate, posted on his personal Facebook on 18 January 2021 regarding the soon-to-be-available COVID-19 vaccine. The general public was encouraged to give their feedback about the soon-available COVID-19 vaccine. The last comment received from the posting was on 27 January 2021. Anti-vaccine sentiments responding to the posting between 18 January and 27 January 2021 were extracted.

### 2.2. Data Analysis

Our data were the comments received on our shared post. The data were copied and pasted manually and imported without participants’ unique identifiers. Textual content from the comments that contain anti-vaccination sentiments was coded. The thematic analysis included reading each comment closely, identifying patterns, assigning codes and formulating themes and sub-themes from the data [6]. To ensure that optimal analytical rigor was practiced, the data were analyzed and coded independently by the researchers (YLW, HA and WLP), after which they were scrutinized, compared and discussed. NVivo 12 software (QSR International Pty Ltd., Doncaster, Australia) was used to conduct a thematic analysis of the comments. The frequency of responses for each theme was recorded.

### 2.3. Ethical Considerations

A note indicating that the comments would be used for research was included in the posting. In addition, it was also noted that all comments provided would be analyzed anonymously in the Facebook posting. This study was approved by the University of Malaya Research Ethics Committee (UM.TNC2/UMREC—1163).

## 3. Results

Between 18 January and 27 January 2021, a total of 1195 responses to the posting were received. Of these, only 276 meaningful textual comments were received. Meaningless mentions and tags were not included in the analyses. Of the 276 mentions, all were anti-vaccine sentiments. An analysis of the comments yielded seven central themes surrounding the COVID-19 anti-vaccine sentiments on issues regarding: (1) long-term side effects and safety; (2) effectiveness; (3) protection duration; (4) the halal status of COVID-19 vaccine; (5) new mRNA technology and vaccine conspiracy theories; (6) low perceived severity; (7) favoring treatment rather than vaccination. Figure 1 illustrates the themes and sub-themes identified in the data. The biggest concerns were about the potential unknown long-term side effect and safety of the new COVID-19 vaccines, and the conspiracy theories around the mRNA vaccine technology.

### 3.1. Long-Term Side Effects and Safety

Worries about the unknown long-term effects of new vaccines were most commonly raised. There were also worries that the side effects may carry on to their children or the next generations. Some felt that the authorities should provide compensation for adverse events following immunization (AEFIs).


*“I want to know if the vaccine has long-term side effects. Why vaccinate everybody in the world? What if the side effect is known after 5 years, all the people in the world will suffer.”*



*“What if the side effect comes later in life? I heard that the effect may be seen in our children or grandchildren, the next generation.”*



*“There has been no large-scale study on the long-term effect of the vaccine on us, such as would it cause cancer, neurobehaviour issues, autoimmune response, allergic reaction, etcetera.”*



*“Imagine if the vaccine recipient is the breadwinner, think of the positive and negative consequences before we take something that may harm our body, the future of our family and descendants.”*



*“Why if there are AEFIs (adverse events following immunization) can’t we sue the government, the vaccine manufacturer? For instance, permanent disability or death….who will be responsible? Aren’t our lives valuable? I suggest COVID-19 vaccine recipients should be given compensation if there are serious side effects following vaccination.”*


Many also feared the safety of the new vaccine. They were concerned about the risks of vaccines that are approved for emergency use. Concerns also stemmed from numerous reports in the media of deaths following vaccination.


*“I am curious, even the existing vaccines also have safety issues, this new vaccine…what guarantee is there that this new vaccine has no safety issues?”*



*“I still doubt this vaccine, because it is still in the experimental phase, and it will have to be improved if there are side effects.”*



*“It is approved under emergency use. The test only ends on 27 January 2023, I hope everyone knows this.”*



*“Low risk? Look at the death of the nurse. Death is low risk? Yes, indeed we refuse this vaccine because of this low risk!”*


### 3.2. Effectiveness

There was mounting concern over the unknown effectiveness of the COVID-19 vaccines, particularly in the event of new SARS-CoV-2 mutations. Some feared that the current COVID-19 vaccines may no longer work to prevent infections of new variants. There was also a concern that all of the evidence of effectiveness was evident from foreign data and that no research has been conducted in the local context that confirms that the COVID-19 vaccines are also effective for people in Asian countries.


*“The COVID-19 vaccines are without sufficient safety and effectiveness. The manufacturers and even the WHO (World Health Organization) stated that the vaccines would not prevent infection, hospitalization or death, only reduce symptoms.”*



*“The WHO (World Health Organization) stated that people can become infected after vaccination. What is more, now that there are new variants. The UK variant of the coronavirus has already been detected in Malaysia.”*



*“I want to know whether the currently established effectiveness also applies to people in Asia. To date, this has only been established in people from the United States and European countries.”*


### 3.3. Duration of Protection

Concern was also raised about the duration of protection of the new COVID-19 vaccine and whether booster shots will be needed.


*“How long will the antibodies last in the body after the second vaccination? Is a booster dose needed ?”*


### 3.4. Halal Status of COVID-19 Vaccine

Questions were raised as to whether the COVID-19 vaccines are halal. Although many agreed upon the permissibility of using vaccines that are not halal under the state of darurah (emergency), there was doubt that the current state has reached the stage of darurah that warrants the use of vaccines that are not yet halal-certified.


*“What is the content of the COVID-19 vaccine? Is it true that there are no non-halal elements or pork DNA? Both science and religion are important, we need a vaccine that is halal as well as not harmful to humans.”*



*“I know non-halal sources can be a must in the state of darurah (a state of dire necessity), but have we reached the darurah stage?”*


### 3.5. New mRNA Technology and Vaccine Conspiracy Theories

Doubts regarding the new messenger RNA (mRNA) vaccine technology used in vaccine production was raised. There were comments that the mRNA vaccines contain microchips that are purportedly implanted in vaccine recipients. Apart from the conspiracy theory that there is a microchip in the COVID-19 vaccine, another vaccine conspiracy theory, such as that the development of vaccines is driven by a profit motive, was brought up.


*“The Pfizer vaccine is a new technology vaccine. Of course, there are concerns because the attempt to use mRNA has failed before, and now it is used in the vaccine.”*



*“I heard about a chip in the vaccine, perhaps it has not been proven. But read this link (a YouTube link attached to the message).”*



*“I heard a chip is inserted in the vaccine and will control our mind, they have planned this a long time ago.”*



*“COVID-19 is not a pandemic and vaccines are a human killer that increases the wealth of some individuals.”*


### 3.6. Low Perceived Severity

There was a low level of perceived severity of COVID-19. Many thought COVID-19 only seriously impacted vulnerable populations.


*“From the statistics, recovery cases are high. As many as 89% of cases are asymptomatic. Our immune system can overcome this disease. Only 11% of people need thorough treatment, the majority are the elderly and people with comorbidities.”*



*“From the beginning of the outbreak, thousands of people in Malaysia have recovered without vaccination. Why do we need a vaccine while many have recovered without vaccination?”*



*“Many people recover from COVID by themselves, what is the importance of vaccination?”*



*“In my opinion, the mass media has created a false sense of the situation, the state we are experiencing now, in my opinion, may not be true because mass media likes to create sensational stories. The data and statistics did not indicate that this outbreak has reached a state of emergency. I just want to emphasize that the mortality rate is low and the risk is only for specific groups.”*


### 3.7. Favoring Treatment Rather than Vaccination

There were also opinions that treatments for COVID-19 are a better alternative than prevention by vaccination. They queried why research does not focus on finding treatment rather than developing vaccination, where there is a need to vaccinate a mass population, and risk the uninfected people. There was also an opinion that vitamin C can be used for the treatment and prevention of COVID-19, which is a much safer alternative.


*“There are few medications that have been found, among these are Ivermectin and hydroxychloroquine. Many peers reviewed have found ivermectin effective as a treatment and prevention. It was found that it can save lives and is safe to use.”*



*“Why is there no treatment found, yet vaccine can be found in such a short time? Why are the experts not researching treatment?”*



*“Vitamin C can also cure COVID-19, there’s no need for vaccine. High doses of vitamin C are effective in reducing death from COVID-19.”*


## 4. Discussion

Malaysia started rolling out its COVID-19 vaccination program in late February 2021. This study was conducted before the availability of COVID-19 vaccines in Malaysia and captured important concerns of the public at a time when COVID-19 vaccines were on the verge of becoming available in the country. Despite the enormous global effort to develop the COVID-19 vaccine, it is not spared from skepticism in Malaysia. This study uncovered a broad range of barriers to COVID-19 vaccine acceptance in Malaysia before the COVID-19 vaccines were available. This study provides information on the extent to which people are open to the new COVID-19 vaccination. The information is useful for health authorities to develop interventions targeted to ease public concerns that may potentially lead to vaccine refusal and hinder the aim of achieving herd immunity through vaccination.

The current unknown long-term vaccine side effects and safety remain one of the greatest challenges to convincing the public to accept the fast-track COVID-19 vaccines. In particular, the family’s sole breadwinner feels insecure about taking the risk of receiving the new vaccine as they fear the financial implications of losing the family’s main income earner. Reports of COVID-19 vaccination causing serious side effects and death in the media have left many fueled with fears about the vaccination. It is well established that COVID-19-related infodemic and disinformation are threatening the acceptance of the vaccine by the public [7]. The public should be made aware that current evidence shows that most of the COVID-19 vaccines are effective and safe, and although they are associated with diverse adverse effects, most of these are minor and resolved quickly, and few result in death [8]. The safety of the vaccine composition should also be highlighted during vaccine promotion [9]. Breadwinners should be the target as they may be the group that exhibits the highest hesitancy toward COVID-19 vaccination. As most of the concern regarding the long-term side effects of the vaccine surrounds the issues of the component of the vaccines, it is of utmost importance to ensure the public about the safety record of the vaccine constituents. The vaccine excipients should be made known to gain the public’s trust in its safety and to address the concern of people who are allergic to certain vaccine constituents; for example, egg proteins.

Media reports of infections after the COVID-19 vaccination left the impression that the vaccine is not effective. The public should be made aware that, although there were reports of infection following COVID-19 vaccination, to date, empirical evidence from nationwide mass vaccination indicates that the risk of testing positive for SARS-CoV-2 after vaccination is low [7]. It is also important to highlight that, when infections do occur in people who have received the COVID-19 vaccine, symptoms tend to be less severe or mild [10]. Concerns have also emerged regarding the possible resistance of SARS-CoV-2 variants to COVID-19 vaccines. Fortunately, recent studies have indicated that the vaccines are protective toward current known COVID-19 variants [11,12], and this should be made known to the public.

*Halal* is a unique Islamic concept and is important in vaccine advocacy in the Muslim community. With Malaysia being a Muslim-majority country, concerns about the vaccines containing substances forbidden by Islam was an ongoing issue before the COVID-19 pandemic. The present study similarly found that the COVID-19 vaccine is also subjected to hesitancy with concerns over whether it is a *halal* vaccine or not. Of note, during the data collection period, the authorities had not officially announced that the use of the COVID-19 vaccine is permissible (*harus*) and obligatory (*wajib*). Currently, in Malaysia, as soon as the vaccination program commenced, it was announced that vaccination against the COVID-19 virus is permissible under the Islamic Rules on the Use of Vaccines. This has greatly reduced the general unease of Muslims who are reticent to accept the vaccine.

Two subthemes emerged under the concern of the new COVID-19 mRNA vaccine. The first concern regarding the mRNA vaccine technology is that it is new and has never been tested on the population at large. The mRNA vaccine is a new type of vaccine and it remains a great challenge for the public to accept a new vaccine whose long-term side effects are as yet unknown [13]. Some benefits of the mRNA vaccine along with its safety have been reported [13,14], and these should be highlighted to gain public confidence. The second concern raised was the link between the new mRNA vaccines and the conspiracy theory. Rumors and conspiracy theories can contribute to vaccine hesitancy. Claims that have been made linking the new mRNA vaccines with microchips being inserted have been widely reported in the media worldwide. Local media and health authorities must help to demystify fake news, disinformation and misinformation about the COVID-19 vaccines.

There is a need to increase the perception of the severity of COVID-19. In the present study, low perception of the severity of COVID-19 is another reason for the opinion that vaccination is unnecessary. The risk of COVID-19 vaccination is presented as more dangerous than the coronavirus disease. It is equally important to enlighten the public that, although the elderly and those with pre-existing conditions are most vulnerable, young people are also at a high risk of contracting and transmitting the virus and should be vaccinated. Cases of young people infecting older family members in shared homes are common. Hence, vaccinating younger people may reduce the risk of infection among the elderly. While the recovery rate from COVID-19 may be high, the long-term health consequence of COVID-19 is unknown. Studies showed that an estimated 80% of the patients aged 17–87 years old that were infected with SARS-CoV-2 developed one or more long-term symptoms [15]. Therefore, the public should be enlightened that, despite high recovery rates among young people, the detrimental after-effects of the infection should not be overlooked. The prevention of infections should be the key priority of the population.

The preference for treatment for COVID-19 versus vaccination was uncovered in this present study. Currently, clinical trials of drugs for the treatment of COVID-19 are ongoing. To date, no medications have yet been proven to be effective in treating COVID-19. On the whole, vaccination is still regarded as one of the most cost-effective healthcare interventions [16]. Although the study of the cost-effectiveness of COVID-19 vaccination is ongoing, a study in the United States showed that influenza vaccination was found to be cost-saving compared to providing treatment [17]. Such information needs to be made known to the public to convince them of the benefit of COVID-19 vaccination compared to COVID-19 therapies. Concerning the perception that the use of high doses of vitamin C may cure the SARS-CoV-2 infection, the public should also be made aware that, currently, there is no solid clinical evidence that supports the use of vitamin C for COVID-19 treatment or prevention. A review of published literature reported that the role of vitamin C in the treatment of patients with SARS-CoV-2 infection was inconclusive and should be further investigated [18].

Of important note, the goal of this study is not to generalize but rather to provide an understanding of people’s reactions to a new COVID-19 vaccine. A major limitation of this study is that we only recorded comments from posts on the researcher’s personal Facebook, so the opinion may not be representative of viewpoints from the general Malaysian population. It is also important to note that the anti-vaccine sentiments found in this study were from virtual communities and do not represent those of communication in the non-social media in Malaysia; therefore, they may not be representative of the general population’s viewpoints. Lastly, generalizability is not typically considered a goal of qualitative research; hence, the quoting of the frequency of responses in the anti-vaccine sentiment themes is merely to allow readers to evaluate the commonness of these themes from the overall responses.

This study was conducted before the COVID-19 vaccines were available. To date, Malaysia is among the most successful nations worldwide in terms of its COVID-19 vaccine rollout. It is being progressively rolled out to different groups based on their risk levels, beginning 24 February 2021. Initially, the Malaysian government recommended healthcare workers and the elderly population as a priority group to receive the COVID-19 vaccines. By 30 December 2021, 97.6% of the adult population in Malaysia were fully vaccinated against COVID-19 [19]. A series of studies have since shown that Malaysians were hesitant to receive COVID-19 vaccines when it was first introduced [20,21]. Likewise, in these studies, concerns about the safety and efficacy of the new COVID-19 vaccines and uncertainty over the new mRNA vaccine technology were paramount [20,22,23]. The Government of Malaysia has made concerted efforts to encourage citizens and non-citizens to register themselves for vaccination via a newly developed application (MySejahtera) [24] as well as outreach programs for rural and interior areas to achieve comprehensive vaccine coverage nationwide [25]. Malaysia’s COVID-19 communication strategy also includes effective communication tackling infodemics, fake news and misinformation about the COVID-19 vaccines to overcome vaccine hesitancy [26]. Further, dine-in and attending religious activities are only allowed for individuals who are fully vaccinated [27]. This may have contributed to the high COVID-19 vaccine uptake in Malaysia. Nonetheless, the perspectives gathered from this study will provide meaningful insights to shape prompt interventions to tackle vaccine hesitancy, particularly during the introduction of new vaccines.

## 5. Conclusions

The present study identified numerous important issues surrounding COVID-19 vaccine mistrust and controversies that may hinder the acceptance of a new vaccine. Mistrust toward COVID-19 vaccines represents a significant challenge in achieving the vaccination coverage rate necessary to achieve herd immunity. COVID-19 vaccination interventions should be equipped to handle the identified concerns found in the study, namely regarding long-term side effects, safety, effectiveness, duration protection, the *halal* status of the vaccine, the use of new mRNA technology in vaccine development, the low perceived severity and the preference for treatment over vaccination. Although the number of new COVID-19 cases worldwide has dropped and many people globally have been fully vaccinated against COVID-19, the anti-vaccine movement persists in some areas of the world. Our findings suggest that efforts to promote COVID-19 vaccine acceptance should focus on counteracting the concerns uncovered in this study. The findings also provide important insight into mitigation strategies surrounding the introduction of new vaccines in the event of future pandemics.

## Figures and Tables

**Figure 1 vaccines-11-00834-f001:**
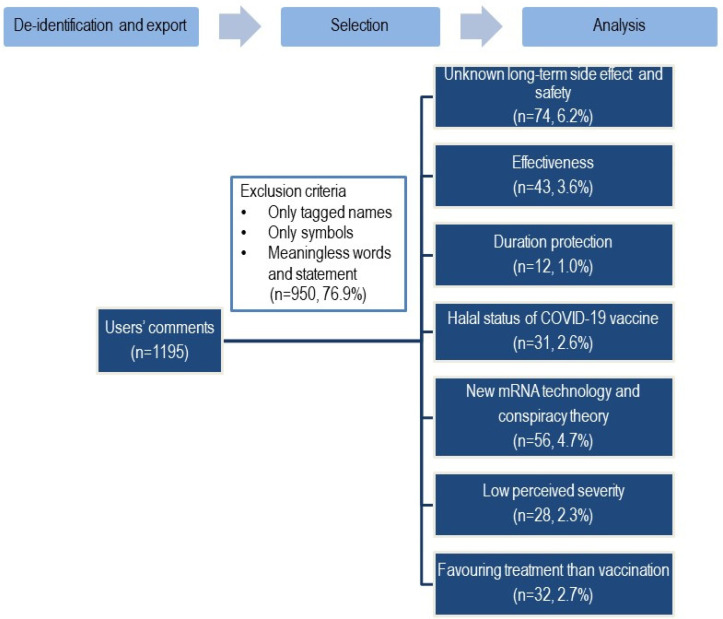
Flowchart showing data collection and uncovered themes surrounding the COVID-19 anti-vaccine sentiments.

## Data Availability

The data presented in this study are available on request from the corresponding author.

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
