# Peer review of "COVID-19 Anti-Vaccine Sentiments in Malaysia: Narratives of Comments from Facebook Post"

_vaccines, 2023, doi:10.3390/vaccines11040834_

Round 1
Reviewer 1 Report
The authors present an interesting work on anti-vaccine sentiments in Malaysa, the article present interesting results, but improvement is needed in parts of the document.
1. I suggest the authors to change the color palette on figure 1 to improve readability and consider the situation of how it will look when printed in grey scale.
2. The authors present: Researcher (MMAHMH), a vaccine advocate, posted on his personal Facebook on 18 January 2021 regarding the soon-to-be available COVID-19 vaccine. The general public was encouraged to give their feedback about the soon-available COVID-19 vaccine. The last comment received from the posting was on 27 January 2021. Anti-vaccine sentiments to the posting between 18 January and 27 January 2021 were extracted.
But more information is needed to evaluate the post, the amount of individual engages on the post, the positive answer against the negatives, is suggested the authors construct a Flow chart showing a summary of the selection process of comments, you can see the work on doi 0.1002/acr.24870 for some ideas on how to build this chart
3. Data was imported without participants’ unique identifiers. How did the authors do this task? Did the authors use an importer tool from Facebook? Was it made by hand? Please specify the method.
4. Analysis of the comments yielded seven central themes surrounding the COVID-19 anti-vaccine sentiments, this part could also be better explained with a flow chart showing the frequency of answer Gruppen in each theme, this will allow the reader to evaluate the comment of the authors and how representative are these themes from the overall answers
5. “This study uncovered a broad range of barriers to COVID-19 vaccine acceptance in Malaysia that may hinder the aim of achieving herd immunity through vaccination” since now should be known the acceptance of the country to vaccination, the authors could cite a work that show how was the vaccine acceptance and discuss how it compares with their analysis
My final comment is that more information on the post and the replies are needed together with a clear presentation of the data and the values in a flow chart, is important for the reviewers and readers to be able to evaluate the information and its quality compared to the engagement of the post, number of replies, positives against negatives sentiments, etc. Also, I will suggest to the authors to upload their anonymized data to an online repository, if review of the original data before processing is needed, I understand a disagreement in this point, and is more a suggestion.
Author Response
Comments and Suggestions for Authors
The authors present an interesting work on anti-vaccine sentiments in Malaysa, the article present interesting results, but improvement is needed in parts of the document.
- I suggest the authors to change the color palette on figure 1 to improve readability and consider the situation of how it will look when printed in grey scale.
Reply: We have removed the former Figure 1, and replaced with a flowchart. Current Figure 1 title is “Flowchart showing data collection and uncovered themes surrounding the COVID-19 anti-vaccine sentiments”
- The authors present: Researcher (MMAHMH), a vaccine advocate, posted on his personal Facebook on 18 January 2021 regarding the soon-to-be available COVID-19 vaccine. The general public was encouraged to give their feedback about the soon-available COVID-19 vaccine. The last comment received from the posting was on 27 January 2021. Anti-vaccine sentiments to the posting between 18 January and 27 January 2021 were extracted.
But more information is needed to evaluate the post, the amount of individual engages on the post, the positive answer against the negatives, is suggested the authors construct a Flow chart showing a summary of the selection process of comments, you can see the work on doi 0.1002/acr.24870 for some ideas on how to build this chart.
Reply: Added Flowchart showing the number of posts for each theme and summary process.
Added in Results section, line 96-99
® Between 18 January and 27 January 2021, a total of 1195 responses to the posting were received. Of these, only 276 meaningful textual comments were received. Meaningless mentions and tags were not included in the analyses. Of the 276 mentions, all were anti-vaccine sentiments.
- Data was imported without participants’ unique identifiers. How did the authors do this task? Did the authors use an importer tool from Facebook? Was it made by hand? Please specify the method.
Reply: Added line 78-79.
® Our data were the comments received on our shared post. The data were copied and pasted manually, and imported without participants’ unique identifiers.
- Analysis of the comments yielded seven central themes surrounding the COVID-19 anti-vaccine sentiments, this part could also be better explained with a flow chart showing the frequency of answer Gruppen in each theme, this will allow the reader to evaluate the comment of the authors and how representative are these themes from the overall answers.
Reply: Added chart and added text regarding the representativeness of the themes.
Also added in Methodology, line 86
® The frequency of responses for each theme was recorded.
And in Results section
® The biggest concerns were about the potential unknown long-term side effect and safety of the new COVID-19 vaccines, and the conspiracy theories around the mRNA vaccine technology.
Added in Limitation, line 313-316
® Lastly, generalizability is not typically considered a goal of qualitative research, hence quoting of frequency of responses in the anti-vaccine sentiment themes is merely to allow readers to evaluate the commonness of these themes from the overall responses.
- “This study uncovered a broad range of barriers to COVID-19 vaccine acceptance in Malaysia that may hinder the aim of achieving herd immunity through vaccination” since now should be known the acceptance of the country to vaccination, the authors could cite a work that show how was the vaccine acceptance and discuss how it compares with their analysis
Reply: Added 324-327
®A series of studies have since shown that Malaysian were hesitant to receive COVID-19 vaccines when it was first introduced [20-21]. Likewise, in these studies, concerns about the safety and efficacy of the new COVID-19 vaccines, and uncertainty over the new mRNA vaccine technology were paramount [20,22,23].
My final comment is that more information on the post and the replies are needed together with a clear presentation of the data and the values in a flow chart, is important for the reviewers and readers to be able to evaluate the information and its quality compared to the engagement of the post, number of replies, positives against negatives sentiments, etc. Also, I will suggest to the authors to upload their anonymized data to an online repository, if review of the original data before processing is needed, I understand a disagreement in this point, and is more a suggestion.
Reply: We added a flowchart, added number to each comments. There were no positive comments were received, which we noted in the manuscript.
We upload our data in the following online repository.
https://kaggle.com/datasets/82458b3cf5b02b11dc9d799e8d3ee8170facbc481290bff8f8a3f8d948823087

Reviewer 2 Report
The paper is really well-written and I can see lot of useful information. The only note, methodology part need to improve and more explanation. However, methodology is clear but I believe that you can provide more information.
Author Response
The paper is really well-written and I can see lot of useful information. The only note, methodology part need to improve and more explanation. However, methodology is clear but I believe that you can provide more information.
Reply: As suggested by reviewer 1, we greatly improved the methodology.

Reviewer 3 Report
Some minor grammatical issues need to be addressed. I particularly draw attention to the use of kosher. In English, this does commonly refer to Jewish law in the same way as halal refers to Muslim religion.
The timing of parts of the paper make it seem outdated and may need to be updated. Alternatively, language that identifies the timeframe is important - eg line 48,49 as well as line 273.

Author Response
Some minor grammatical issues need to be addressed. I particularly draw attention to the use of kosher. In English, this does commonly refer to Jewish law in the same way as halal refers to Muslim religion.
Reply: We proofread the manuscript again and made a considerable corrections over the grammatical mistakes. We remove the word kosher in the entire manuscript.
The timing of parts of the paper make it seem outdated and may need to be updated. Alternatively, language that identifies the timeframe is important - eg line 48,49 as well as line 273.
`Reply: Updated line 50-51
®On December 11, 2020, the Food and Drug Administration (FDA) has authorized the first COVID-19 vaccine for emergency use.
Added in Discussion line 317-337
®This study was conducted before the COVID-19 vaccines were available. To date, Malaysia is among the most successful nations worldwide in terms of its COVID-19 vaccine rollout. It is being progressively rolled out to different groups based on their risk levels beginning 24 February 2021. Initially, the Malaysian government recommended healthcare workers and the elderly population as a priority group to receive the COVID-19 vaccines. By 30th December 2021, 97.6% of the adult population in Malaysia were fully vaccinated against COVID-19 [19]. A series of studies have since shown that Malaysian were hesitant to receive COVID-19 vaccines when it was first introduced [20-21]. Likewise, in these studies, concerns about the safety and efficacy of the new COVID-19 vaccines, and uncertainty over the new mRNA vaccine technology were paramount [20,22,23]. The Government of Malaysia has made concerted efforts to encourage citizens and non-citizens to register themselves for vaccination via a newly developed application (MySejahtera) [24] as well as outreach programs for rural and interior areas to achieve comprehensive vaccine coverage nationwide [25]. Malaysia's COVID-19 communication strategy also includes effective communication tackling infodemics, fake news, and misinformation about the COVID-19 vaccines to overcome vaccine hesitancy [26]. Further, dine-in and attending religious activities are only allowed for individuals who are fully-vaccinated [27].. This may have contributed to the high COVID-19 vaccine uptake in Malaysia. Nonetheless, the perspectives gathered from this study will provide meaningful insights to shape prompt interventions to tackle vaccine hesitancy, particularly during the introduction of new vaccines.

Round 2
Reviewer 1 Report
First, I want to thank the authors for answering my questions, I don't have more comments.